# The Role of Fascial Tissue Layer in Electric Signal Transmission from the Forearm Musculature to the Cutaneous Layer as a Possibility for Increased Signal Strength in Myoelectric Forearm Exoprosthesis Development

**DOI:** 10.3390/bioengineering10030319

**Published:** 2023-03-02

**Authors:** Mark-Edward Pogarasteanu, Marius Moga, Adrian Barbilian, George Avram, Monica Dascalu, Eduard Franti, Nicolae Gheorghiu, Cosmin Moldovan, Elena Rusu, Razvan Adam, Carmen Orban

**Affiliations:** 1Faculty of Medicine, “Carol Davila” University of Medicine and Pharmacy, 8 Eroii Sanitari Boulevard, 050474 Bucharest, Romania; 2Department of Orthopaedics and Trauma Surgery, “Dr. Carol Davila” Central Military Emergency University Hospital, 010242 Bucharest, Romania; 3Faculty of Electronics, Telecommunications and Information Technology, University Politehnica of Bucharest, 313 Splaiul Independentei, 060042 Bucharest, Romania; 4Center for New Electronic Architecture, Romanian Academy Center for Artificial Intelligence, 13 September Blulevard, 050711 Bucharest, Romania; 5Microsystems in Biomedical and Environmental Applications Laboratory, National Institute for Research and Development in Microtechnology, 126A Erou Iancu Nicolae Street, 077190 Bucharest, Romania; 6Department of Orthopedics and Traumatology, Elias Emergency University Hospital, 011461 Bucharest, Romania; 7Department of Medical-Clinical Disciplines, Faculty of Medicine, “Titu Maiorescu” University of Bucharest, 031593 Bucharest, Romania; 8Department of General Surgery, Witting Clinical Hospital, 010243 Bucharest, Romania; 9Department of Preclinic Disciplines, Faculty of Medicine, “Titu Maiorescu” University of Bucharest, 031593 Bucharest, Romania; 10Department of First Aid and Disaster Medicine, Faculty of Medicine, “Titu Maiorescu” University of Bucharest, 040051 Bucharest, Romania; 11Department of Anesthesia and Intensive Care, “Carol Davila” University of Medicine and Pharmacy, 020021 Bucharest, Romania

**Keywords:** fascia, myoelectric signals, exoprosthesis, forearm

## Abstract

Myoelectric exoprostheses serve to aid in the everyday activities of patients with forearm or hand amputations. While electrical signals are known key factors controlling exoprosthesis, little is known about how we can improve their transmission strength from the forearm muscles as to obtain better sEMG. The purpose of this study is to evaluate the role of the forearm fascial layer in transmitting myoelectrical current. We examined the sEMG signals in three individual muscles, each from six healthy forearms (Group 1) and six amputation stumps (Group 2), along with their complete biometric characteristics. Following the tests, one patient underwent a circumferential osteoneuromuscular stump revision surgery (CONM) that also involved partial removal of fascia and subcutaneous fat in the amputation stump, with re-testing after complete healing. In group 1, we obtained a stronger sEMG signal than in Group 2. In the CONM case, after surgery, the patient’s data suggest that the removal of fascia, alongside the fibrotic and subcutaneous fat tissue, generates a stronger sEMG signal. Therefore, a reduction in the fascial layer, especially if accompanied by a reduction of the subcutaneous fat layer may prove significant for improving the strength of sEMG signals used in the control of modern exoprosthetics.

## 1. Introduction

Progress in the field of forearm and hand prosthetics has been astonishing in recent years, seeing improvement in both aesthetics and in the functionality of robotic forearm and hand prostheses [1,2]. Surgical techniques of amputation and amputation stump management have also evolved, increasingly focusing on techniques of reinnervation and re-distribution of the biological control system to make use of electrical signals from the brain that are devoid of effectors [3,4,5].

In 2013, the Circumferential OsteoNeuroMyoplasty (CONM) amputation method was registered [6,7,8], aiming to improve the methodology of amputation of a limb by suturing the muscle groups in such a way as to superficialize the deep muscle groups, allowing better and distinct control of the movement of the elements of an exoprosthesis worn by the patient. The CONM methodology allows fasciectomy at the amputation stump as well as reduction of the subcutaneous layer to reduce the electrical resistance of the tissue layer between the muscle masses and the surface electrodes, thus amplifying the signal collected by the surface electrodes.

EMG signals, representing the graphic recording of the electrical potential emitted by muscle cells, are used for controlling myoprostheses [9]; it should be noted that the thickness of the fat layer between the muscle and the EMG sensor decreases the signal intensity in direct proportion [10]. The attenuation of the electrical signal emitted by the muscles in their contraction seems to depend on the distance from the muscle to the surface electrode, the placement of the electrode with respect to the muscle, as well as on the thickness and structure of the layers that are interposed between them [11,12]. This may be studied in either a clinical scenario, animal model or in a three-layer finite element muscle model, consisting of skin, fat and muscle, as the main layers [13]. A 2013 study by Minetto discovered that the amplitude and mean frequency of the sEMG signals increase with the decrease in the subcutaneous tissue thickness [14]. Another aspect that may affect the intensity of the sEMG signal is the movement of each muscle, with a significant effect [15]. Subcutaneous tissue and fascia location, as well a scar tissue, may be evaluated by means of ultrasonography; optimally, the sEMG sensors are placed as close as possible to the targeted muscle masses [6,14,16,17]. Moreover, the “background noise” (electrical fields produced by the other muscles in close proximity) can further add difficulty in a clear signal detection [18]. This process can be explained by the scattering of the weaker electrical signal through several structures to the surface and thus overlapping with other signals, a problem that may be addressed by using sEMG pattern recognition techniques [9].

The concept of implanting EMG sensors goes in two main directions: the use of sensors directly connected to the exoprosthesis, basically percutaneously implanted intramuscular electrodes, and the use of sensors that are surgically placed in the muscle mass, that are autonomous, communicating with the exoprosthesis via wireless technology [19,20]. As far as percutaneously implanted electrodes are concerned, the method has been less used than surface electrodes, although one study [21] establishes their clear advantage over sEMG electrodes.

A number of recent studies have proposed electrode implantation in the hope of retaining the advantages of percutaneously implantable electrodes (real-time measurement accuracy) and eliminating the disadvantages (risk of infection and difficulty in keeping the electrode implanted and attached to the prosthesis simultaneously over the long term) [22]. The 2010 research of Culjat J. et al. [23] shows a myoelectric sensor implanted in forearm amputation stumps and tested using a myoelectric prosthesis. Success is reported in the wireless reception and transmission of the signal from the muscle, as well as the practical control of the prosthesis, which patients find more intuitive than the classical control option. Pasquina et al. [24] in 2015 presented a subcutaneous implantable electrode, with a surface communication zone, made of titanium and having a discoid shape, implanted in one of the authors of the study. Thus, although not fully implanted in the body, this implant deserves attention as an intermediate form between percutaneously implanted electrodes and fully implantable forms. Tests revealed the superiority of the method over the use of surface electrodes, with the signal received from the implant having superior electrical properties to that received by sEMG and being less influenced by mechanical interference.

The purpose of this study is to evaluate the differences in subcutaneous and fascia layers between forearm amputation stumps and non-amputated forearms and forearms. The hypothesis is that the amputation stumps have different anatomical and bioelectrical characteristics when compared to non-amputated forearms. The main objective is to measure the anatomic and bioelectric characteristics of the forearm and hand amputation stumps and to compare them to the corresponding measured characteristics of non-amputated forearms. The first secondary objective is to implement a virtual reality augmented training protocol for subjects with forearm and/or hand amputation that will allow them to use an intelligent bionic prosthesis with myoelectric control system created by the Polytechnic Institute Bucharest team, with a background in the development of such technical solutions [25,26]. The second secondary objective is to present the case of one test subject that underwent the CONM procedure during testing.

This study is a case series, as the small number of test subjects does not allow for a pilot study.

## 2. Materials and Methods

### 2.1. Test Subject Selection

The Polytechnic Institute Bucharest team’s background in the development of myoelectric prosthesis allowed access to an internal database of amputee and non-amputee test subjects that were contacted in the process of recruitment and informed of the study protocol, the duration, the inclusion and exclusion criteria and the potential risks. Of those interested in participating, 5 test subjects were selected who had a prior forearm or hand amputation and one who had no forearm or hand amputations.

The inclusion criteria were unilateral or bilateral amputation of the forearm or of the hand, age over 18 years and the amputation was performed over 6 months prior to the tests. The exclusion criteria were insufficiently matured amputation stump (under 6 months from surgery, exhibiting pain, edema or phantom limb pain), amputation of just one or of several fingers, local pathology at the level of the amputation stump that would make picking up EMG signals difficult or impossible (such as scarring, local infections and hypersensitivity) and associated pathologies that may impede the cooperation between researcher and test subjects for the virtual reality training (such as mental disorders or visual impairment).

In the case of one test subject, the amputation was bilateral, so there were in fact 6 forearm amputation stumps tested. The test subjects were assigned to two groups. Group 1 comprised of 6 amputated forearms and Group 2 of six non-amputated forearms, specifically 4 from subjects that had their other forearms amputated and two from a healthy participant. Their general characteristics were determined as such: the level of amputation, the side on which the amputation was performed (right or left), the time passed from the amputation, the surgical indication for the procedure (trauma, tumor, infection, thermal injury), the type of amputation as well as the existence of keloid scarring.

### 2.2. sEMG Measurements

A series of measurements were performed about both Group 1 as well as Group 2, to the extent possible:(a)Measurement of anthropometric and kinematic data—the largest circumference in the proximal third of the forearm, as well as in the middle and distal third, adipose layer thickness in the proximal, middle and distal third of the distal forearm, elbow flexion deficit, elbow extension deficit and the degree of prono-supination. The adipose layer was measured using standard calipers, and the flexion, extension, pronation and supination angles were determined using a standard medical goniometer.(b)Recording of EMG data—both the amputation stumps as well as the healthy forearms were tested using a standard EMG for the retrieval of surface-level myoelectric signals; two models were used: Alpin Biomed Keypoint 4 and Anjue CMS-6600. The initial stage, that of measuring the EMG signals, was conducted for all 6 participants in the study, both the amputated forearms as well as the healthy ones. The muscles that contribute to the major movement of the forearm and of the hand were selected for recording of the EMG data, one muscle representing each of the three compartments of the forearm: the flexor carpi radialis muscle from the anterior compartment, in the superficial plane and innervated by the radial nerve; the brahioradialis muscle from the lateral compartment, in the superficial plane and innervated by the radial nerve; and the extensor digitorum communis muscle, in the superficial plane, innervated by the radial nerve.(c)Skin preparation for both the amputated and non-amputated limbs—where it was needed, the amputation stump was cleaned of excessive hair follicles, if these prevented the surface EMG electrode from adhering to the skin, then a 72% alcoholic solution was used locally to remove excess sebum. The electrodes were applied to the surface of the amputation stumps, as well as to the unaffected forearms, in corresponding areas, to pick up the signals emitted by the same muscle. For the test subject with the bilateral amputation, both stumps were tested, while the healthy subject was tested for both unaffected forearms.(d)Data acquisition and processing of the EMG signal—the subjects performed voluntary isometric contractions, with 1 s intervals, for 30 to 60 s, each in accordance with their individual capacity for effort (Figure 1). The signals were imported in MathWorks MatLab software for processing (https://www.mathworks.com/products/matlab.html (accessed on 5 January 2023)) to eliminate the acquisition errors and to highlight the differences between the first set of data (Group 1, the amputation stumps) and the second set of data (Group 2, the unaffected forearms). Using the Biopac Systems AcqKnowledge software (https://www.biopac.com/product/acqknowledge-software/ (accessed on 5 January 2023)), the mean value of the amplitudes in effective contraction intervals were calculated, where an effective contraction interval is defined as the interval of voluntary muscle contraction that produces a distinct signal that is strong enough to be picked up by surface EMG sensors.

### 2.3. Signal Processing

The AcqKnowledge software package was used for the acquisition and processing of the signals, with the purpose of calibrating them, so that they may be used in the command of a myoelectric prosthesis’ electrical circuits and mechatronic components. This software package allows for viewing, measuring, transforming, replaying and analyzing data and is produced by Biopac Inc. (https://www.biopac.com/product/acqknowledge-software/ (accessed on 5 January 2023)). After the employment of several functions (such as the envelope function and filters) the wave signals were processed, obtaining a clear form, with no over imposed background noise; this signal is compared to a threshold, to translate the signal emitted by the muscle and to translate it into a rectangular signal that may be used by the electronic system. We note that the processing of the signal has a short time of action, the delay caused by this between the moment of sEMG signal acquisition and the command emitted by the electromecathronic signal being unnoticeable to the patient. The application of specific software processing filters makes it possible to obtain, from an sEMG signal that is difficult to distinguish from “background noise” and difficult to use, a series of simplified data that can easily be used to control the experimental bionic hand prosthesis model.

After surface EMG measurements were performed, individualized training of the flexor carpi radialis, brahioradialis and extensor digitorum communis muscles commenced, consisting in virtual environment assisted muscular contractions, in which 3 of the 5 patients participated, with a total of 4 amputation stumps. As far as the training sessions are concerned, they will consist of muscle contractions performed by the patient, both at the level of the stump and at the healthy limb, following a predetermined protocol, with the display of information collected by EMG sensors in virtual format, or the direct translation of these movements into movements of the exoprosthesis.

### 2.4. The Virtual Reality Training Protocol

A training protocol that involves the healthy, non-amputee limbs was chosen. The subjects performed the same gestures and attempted to contract the same muscle groups simultaneously, with the aid of a visual representation as control. The training protocol is based on a working system incorporating both software and hardware elements and can be easily adapted even for use by subjects with double forearm amputation. Healthy hand movements are monitored and entered in a computer system using a physical–virtual reality interface solution called The Virtual Reality Glove (DG5 VHand, produced by DGTech Engineering Solutions, 2600 South Shore Boulevard, League City, TX 77573, USA). This solution registers the input signals and replicates in the virtual environment the movements performed by the test subject, depositing the output algorithms in a database [24]. This will allow the medical team to mirror the gestures during training, which will then be systematized into movement patterns.

### 2.5. sEMG Interface

The software elements consist of the totality of the programs that make up the virtual environment for configuration, testing and graphical representation, while the hardware components consist of the virtual reality glove, connectivity elements, processing elements and the experimental myoelectric prosthesis model, which connects to the amputation stump via an sEMG interface. The sEMG interface is mounted on the amputation stump and is connected to the experimental bionic hand. The patient is connected to the software system via the virtual reality glove in cases of healthy limbs, and via the exoprosthesis or a set of sEMG sensors in cases of amputation stumps. The software system displays the virtual counterpart of the two limbs (healthy and amputated) also in the form of the forearm and hand to facilitate patient interaction with the system and to facilitate visual feedback (Figure 2).

Training in the virtual environment consists, after familiarizing the patient with the working environment and the equipment, in performing synchronous movements of the two thoracic limbs, represented on the screen as two hands, the information coming from the interpretation of the sEMG signals on the one hand and the signals transmitted by the virtual reality glove on the other hand. The focus is on performing precise movements with the amputated hand, guided by the simultaneous performance of the movements and the visual feedback obtained from the movements performed by the graphical limb representation.

Thus, the virtual environment initially functions as an abstract training environment, and as the patient learns to better control muscle contractions at the amputation site and perform more precise movements of the corresponding virtual hand, these commands are transmitted to the experimental exoprosthetic model, which will in turn provide visual confirmation of movement performance.

### 2.6. The Experimental Myoprosthesis

Finally, patients are trained directly with the experimental exoprosthesis as a preparatory step for the use of a commercial myoelectric prostheses (Figure 3).

### 2.7. Virtual Reality Components

The virtual working environment allows for testing multiple algorithms for complex motion, summing characteristics of the decomposed motions executed by the patient in an individualized pattern. The P5 Glove was used to receive data from the unaffected side and upload them online. The glove registers signals and replicates in the virtual environment the movements done by the research subject, depositing the algorithms in a database. The virtual environment component has a modular structure, each module with a specific function: generating the virtual action environments, description of the artificial virtual hands, description of the myoprosthesis, description of the control program, coordination and synchronization.

The Command-and-Control Module has the function of receiving the information from the virtual reality glove and the sEMG sensors, processing the information according to predetermined algorithms and transmitting the results to both the exoprosthesis, as well as to its correspondence in the virtual environment. The Graphic Interface allows for the visual feedback necessary to the patient during each training session, in order to observe the correlation between the movement of the hands in the virtual environment and the movement of the hands in the real-life setting, which will allow for a finer control over the musculature in the amputation stump. The graphic interface displays the following joints: humero-ulno-radial (elbow), radio-carpal (wrist), carpometacarpal, metacarpophalangeal and interphalangeal.

### 2.8. Signal Modulation

To eliminate additional signals that may interfere with the sEMG waveforms, an instrumentation amplifier was used. After amplification, the signals are filtered, keeping only those in the 20 Hz–450 Hz range. This eliminates the signal component produced by the membrane potential and interference from electrode contact with the tegument and micro-movements at this level. The movable elements of the experimental prosthesis are equipped with Pulse Width Modulation (PWM) controlled servomotors, sensors located at the tactile area of predilection (area corresponding to the distal phalanx). The sensor signals are first amplified, then processed and transmitted to the control block for analysis of the pressure level exerted by the prosthesis on the objects manipulated under direct patient command. Thus, control of the experimental prosthesis can be achieved at the level of each finger, using sEMG signals received from the forearm muscles.

### 2.9. The Individual Training Sessions

The individual training of each subject consisted initially of simple execution of flexion and extension movements at the finger level, initially with just one finger, then multiple fingers simultaneously. The movement was reproduced at the virtual reality level by receiving, processing and interpreting the signals received from the virtual reality glove and the sEMG sensors placed at the amputation stump. Each subject was asked to focus on precise movement performance, with the major goal of achieving better control during fine movements, empirically determined by the patient. Following initial training, rapidly improved control of the prosthesis basic functions was observed, and subjects were asked to manipulate objects using the experimental myoprosthesis.

## 3. Results

### 3.1. General Subject Description

The general characteristics of the amputation stumps are presented in Table 1 while the characteristics of the test subjects are presented in Table 2.

### 3.2. Anatomical Characteristics

The anatomical characteristics determined from direct measurements of healthy forearms and amputation stumps are summarized in Table 3 and Table 4.

### 3.3. Mean sEMG Amplitude Values

Table 5 presents the results on the sEMG signal amplitude of flexor carpi radialis, brachioradialis and extensor digitorum communis muscles over the effective contraction interval.

### 3.4. The CONM Patient Data and Procedure

Following the amputation, and after the wound healing period, one of the patients had sEMG signals collected from the amputation before re-entry into training (Figure 4).

During the individual training sessions, one of the subjects reported experiencing chronic pain at the level of the amputation stump, linked to a series of muscle retractions and bone spurs. The subject underwent a surgical procedure to revise the stump through the CONM technique, independent of the study itself (Figure 5).

As required by the CONM technique, a preoperative radiological study was performed to thoroughly evaluate the level of the needed bone spur ostectomy and radial and ulnar osteotomy. The stump was revised by a subcutaneous dissection and formed two flaps and visualization of the antebrachial fascia, which was later partially removed, together with approximately 20–30% of the subcutaneous fat, in a circumferential manner. Each muscle and each nerve was individually dissected and identified, while the radius and ulna were trimmed of bone spurs and shortening osteotomy was performed on each of them. The next step involved drilling transosseous holes to anchor the individual muscles. Each individually dissected muscle was brought in the same plane and sutured to the radius and ulna, while the three nerves were transposed superficially. A single layer suture was performed, followed by a sterile dressing and a post-operative X-ray to evaluate the new resection level of the forearm bones. The healing period was uneventful, and after three months the patient was re-included in the individual virtual reality assisted training sessions (Figure 6).

## 4. Discussion

Anatomical measurements of the amputation stumps showed that they tended to have a reduced circumference and a slightly reduced fat layer. These tendencies were also observed visually, both by the subjects and the examiners, and can be interpreted in the context of muscle atrophy installed over time. This information correlates with sEMG measurements, revealing severe muscle atrophy. This is relevant, as Jiang et al. in their 2012 article [27] found that the degrees of freedom of a transradial amputee can be estimated using surface EMG of the remaining muscles in the amputation stump. The direct measurement by calipers is an alternative to ultrasound and/or MRI volumetric measurements [28], with an added advantage of being cheap and fast, but presumably less reliable than an MRI measurement. Muscle atrophy alters muscle morphology, which may play a role in sensor positioning and sensor number, with a direct impact on in vivo use of myoelectric prosthesis, as found by Naik et al. [29].

Regarding flexion and extension at the elbow level, as well as pronation and supination at the level of the remaining stump or at the level of healthy forearms, we did not observe significant deficits, interpreting this as a consequence of the patients’ use of aesthetic or mechanical prostheses on a routine basis, which resulted in the maintenance of a degree of mobility, although the muscles suffered an advanced degree of atrophy. Geng et al. [30] found that the variation of the position of a subject’s arm has a direct impact on the control of a myoprosthesis; thus, any impact on the degrees of motion that the subject is able to execute we believe to be relevant. Better results in controlling a myoelectric prosthesis were obtained when extrinsic data regarding the amputation stump’s spatial positioning were combined with sEMG data [31]; thus, it may be desirable to maintain as much amplitude of motion concerning the remaining joints of the amputated side as possible. Moreover, EMG data alone may be used for mirroring-like control of a myoelectric prosthesis, using data from the unaffected side [32], impacted by the available range of motion, suggesting that maintaining an optimal range of motion of the unaffected side may also be relevant.

We found a remarkable difference between the two sets of sEMG signal values collected from amputation stumps and healthy forearms, a difference due to muscle atrophy and fibrosis in the amputation stump, which can be approximated to two orders of magnitude. In each case, and for each of the muscles chosen, the values experienced low variability between the data in each set.

We observed that the surface EMG signals collected from the amputation stumps (Group 1) had a decreased amplitude when opposed to those collected from the healthy forearms (Group 2). This was observed to be the case both for each patient that presented the association amputated forearm—healthy forearm, as well as in the global comparison of the two groups, which included a double amputee and a healthy subject. To overcome this difficulty and to perform an adequate analysis of the signals, a software manipulation was performed on the data from Group 1 in order to remove background noise signals and to extract a clear set of data to be compared to the data from Group 2, as well as to be used by the software that controls the virtual environment. We did not consider a variability in sampling rates, which may be relevant in future testing, as Li’s 2010 article would suggest [33]. Further improvement in the utility of the data could have been obtained by combining with EEG (electroencephalography) data, as suggested by Li [34], but in our opinion this would be impractical in a clinical setting. Other authors [35,36] dispute the importance of sEMG signal amplitude, when compared to the patterns that emerge for each patient, patterns that seem to be different for forearm amputees when compared to normal limbed test subjects, and are also dependent on the distribution of the electrodes. In further studies, amplitude is not the only factor taken into consideration, but synergistic firing of muscles as well [37], while further studies try to establish a suitable classifier to aid in discerning the right type of data to be extracted from the test subjects for optimal myoelectric prosthesis control [38].

An increase in amplitude value is noted in the case of the subject who underwent the surgical intervention of amputation stump retouching, which suggests (in the context of the impossibility of extrapolation from a single case) the usefulness of the CONM technique in improving the qualities of the sEMG signal received from the amputation stumps. As an interpretative difficulty, we note that the three muscles chosen (flexor carpi radialis, brachioradialis and extensor digitorum communis) following revision were transpositioned and measurements were performed at the new locations. In this case, the possibility of intraoperative misidentification of each muscle mass arises; this can be avoided by careful dissection, and we postulate that accurate identification of each muscle is secondary, the important thing being the correct execution of the operative technique, which ensures transposition not necessarily of all muscles absolutely, but mainly of those that have lost as little mass as possible through atrophy. In other words, it seems more useful to transpose a deep, voluminous muscle to the superficial plane than to keep a muscle that has atrophied but corresponds to the area of skin projection in the superficial plane. The CONM technique is one of the few modern-day surgical innovations in the field of forearm amputations, as most techniques focus on targeted reinnervation and osseointegration [39], while the main focus of research is on the exoprosthesis itself [40] and on the interface between the amputation stump and the exoprosthesis [41].

One difficulty encountered in the sEMG data collection process is achieving accurate contraction of only one muscle, especially as regards amputation stumps. Of course, a solution would have been to collect EMG signals via an intramuscular probe, an electrode that is implanted, but this would have changed the nature of the research. Moreover, muscle signal strength could have been augmented via targeted muscular reinnervation [42,43], but this also would have gone beyond the scope of the study. Another difficulty is in receiving a clear signal, distinct from those around it. We accept that the received data may be altered by adjacent sEMG signals and patient arm position [44,45], which is why we chose to use a single signal from each compartment, namely the strongest and clearest signal from a superficial muscle, and this determination was made empirically, by repeated serial determinations on both healthy forearms and amputation stumps. Moreover, there is the challenge of interpreting the data in a practical context, as amputations were performed by the standard technique, which implies the possibility of muscle retraction, if the anchoring of muscle groups to each other over the severed bone ends was not done meticulously. More recently, EMG data are being interpreted not only in direct per-signal approach, as our paper presents, but also in patterns of muscle activation, representing patterns of motion of the now-amputated limb [46,47,48]. These patterns [49] may not be modified through the CONM method, or through any amputation method that allows for a change in the disposition of the forearm musculature, as the firings of the muscles are based on a neurological command, not on their physical location, but they may be modified through targeted muscle reinnervation, still allowing for exoprosthesis control [50]. Other issues have been reported with EMG signal conditioning and sampling, such as difficulties with cut-off frequency and a low-sampling rate [16], as well as mechanical difficulties, such as stump mobility interfering with sensor placement through secondary displacement [51]. Although pattern recognition seems a more promising direction, a 2018 study by Resnik et al. found direct myoelectric control to be slightly superior to pattern recognition in a cross-over study [52].

The use of the experimental myoprosthesis offered all the advantages of a commercial exoprosthesis, having the possibility of being incorporated into a sleeve and used by the subjects, but also offered the advantage of the possibility of directly studying the process of creation of this type of myoprosthesis. Thus, it was possible to study the process of collecting sEMG signals, as well as the process of training subjects in the use of this myoprosthesis, using a software environment.

Using a software environment is not singular [53,54] and offers the possibility of testing the subject’s ability to voluntarily control the musculature at the amputation site and, following the series of tests, choosing the subset of sEMG signals most useful in the use of a myoelectric prosthesis. Moreover, using the virtual equivalents of the healthy hand and exoprosthesis allows the subject to become familiar with this type of myoprosthesis and train safely before the prosthesis is purchased. The subject is given the possibility to interact with the software application through the acquisition module in order to perform predefined sets of movements initially in the virtual environment, which can then be translated into the physical environment through gestures that can be integrated into the experimental myoprosthesis; thus, the subject is given the possibility that instead of commanding simple movements through an sEMG signal, he/she can determine the performance of complex movements, specific to the subject. The subject’s control over the myoprosthesis can be tested and optimized by adjusting the sensor arrangement and training type for maximum movement freedom. The exoprosthesis movements can be calibrated by analyzing virtual movements for better mechanical alignment and maximum function for each subject. Myoprostheses are currently used extensively as a research model in exoprostheses, as they offer a good number of functions, while providing the test subject with a comfortable experience [55].

The process of training the subjects in the chosen virtual environment system (one of many described [56]) involved the use of a myoprosthesis in conjunction with the use of a virtual working environment, in which a graphic representation of their limbs was projected, performing the same movements as the subject. This visual feedback, in conjunction with tactile information gathered by sensors at the level of the prosthesis and translated into visual information (percentage of intensity on a visualization bar) allowed the initial phase of training, which had been anticipated at the beginning of the research, to be overcome, namely the simple control of fine movements, and the performance of complex movements, such as manipulating fragile objects. This suggests that the cumulative effect of the techniques used is remarkable.

Just as Teklemariam A. et al. [11] highlighted in his paper, our study also concludes that the main biological materials shielding the electrical impulse from the source (muscle) to the surface electrode are the subcutaneous layer together with the fascia. This is equally true for both the classical trans-radial amputation technique and for the CONM technique.

### Limitations of the Current Study

Statistical interpretation of the data was not performed due to the small number of subjects. Moreover, gender influence was not analyzed, as the test subjects were all male. Data stratification according to age was not performed, due to the small size of the sample. CONM was performed incidentally, and on only one subject, so its influence on the resulting data cannot be extrapolated to all the subjects.

## 5. Conclusions

sEMG signals collected from amputation stumps are qualitatively inferior to those collected from healthy forearms, but still usable in controlling an experimental model of myoprothesis with sensory interface.

The quality of sEMG signals received from the surface of an amputation stump can be improved by using the CONM surgical technique, which consists, among other surgical steps, of partially removing fascia and subcutaneous fatty tissue.

Training the patients using the experimental myoprosthesis model, together with the use of virtual reality as well as the sensory interface, resulted in an improved level of control for commanded movements in the amputation stump.

We can conclude that the main biological materials shielding the electrical impulse from the source (muscles) to the surface electrode are the fascia together with the adipose tissue, and these can be modified with surgery.

## Figures and Tables

**Figure 1 bioengineering-10-00319-f001:**
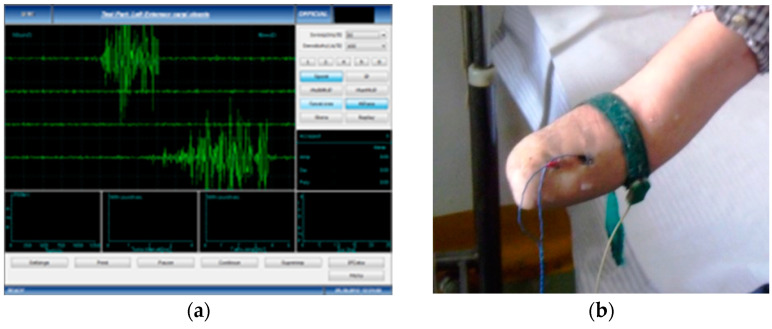
Harvesting the sEMG signals: (**a**) Dedicated software for accurate measuring; (**b**) Placement of the measuring electrodes, after correct skin preparation.

**Figure 2 bioengineering-10-00319-f002:**
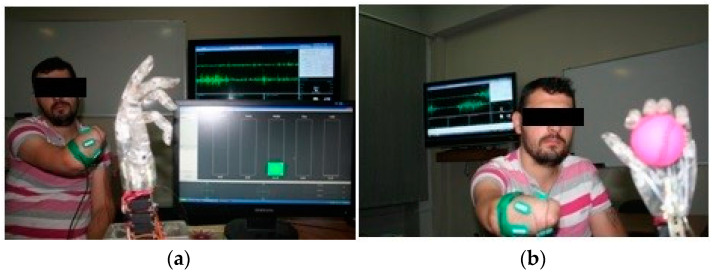
Testing the amputee patient with the myoelectric hand: (**a**) The myoelectric hand in resting position; (**b**) The myoelectric hand during an active movement, connected to the sEMG receiver.

**Figure 3 bioengineering-10-00319-f003:**
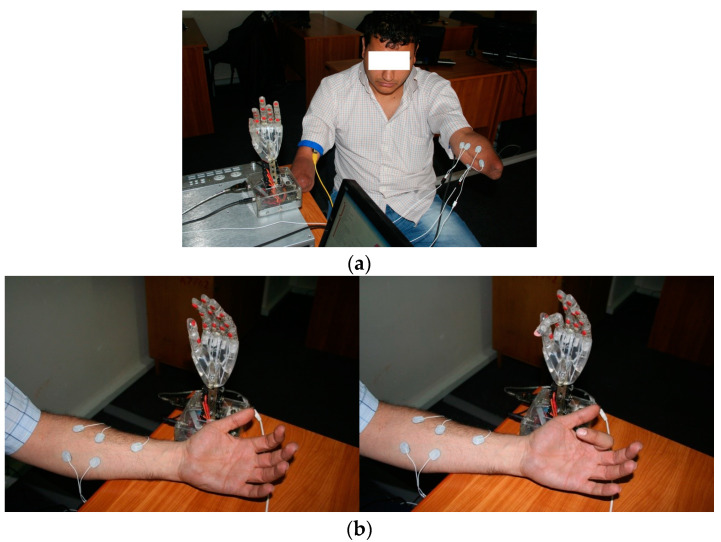
Patient training directly with the experimental exoprosthesis: (**a**) Double-amputated subject during preliminary training using direct connection with experimental exoprosthesis; (**b**) Bionic hand exoprosthesis used by an indemnified subject, in static position and during active movement.

**Figure 4 bioengineering-10-00319-f004:**
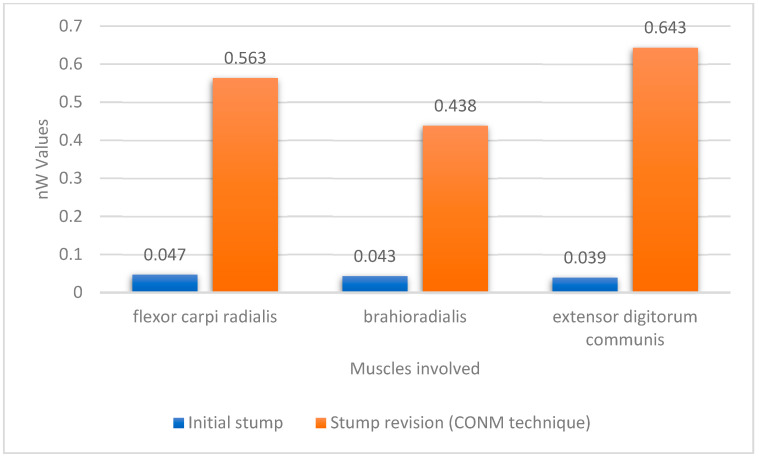
Plot of the mean amplitude values for the sEMG signals before and after the CONM technique blunt revision.

**Figure 5 bioengineering-10-00319-f005:**
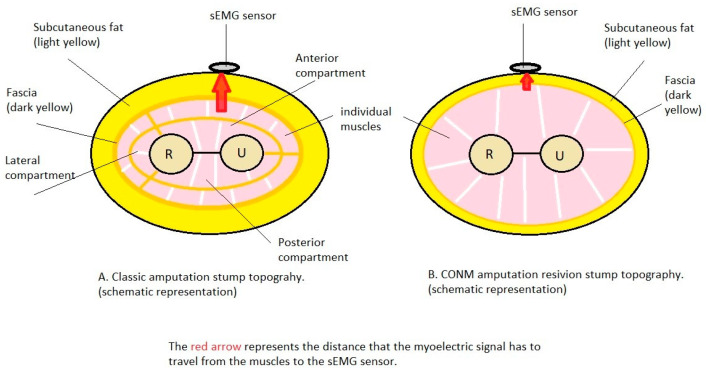
The CONM surgical technique with partial excision of the forearm fascia together with a reduction in the subcutaneous fat layer and re-disposition of the individual muscles.

**Figure 6 bioengineering-10-00319-f006:**
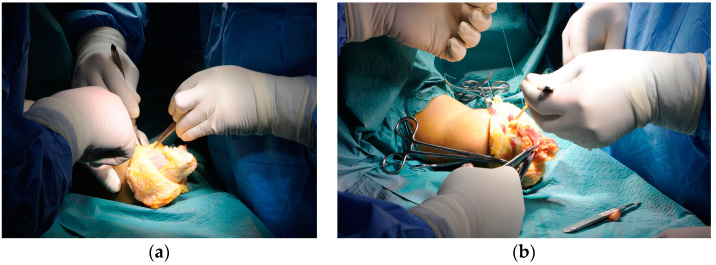
A live intraoperative view of the CONM technique: (**a**) Partial excision of the forearm fascia; (**b**) circumferential transposition of the muscle groups.

**Table 1 bioengineering-10-00319-t001:** The general characteristics of the amputation stumps.

Stump No.	Subject	Amputation Level	Amputation Side	Time from the Amputation	Surgical Indication	Type of Amputation	Posttraumatic Scar
1	F.B.	forearm proximal 1/3	right	5 years	trauma	standard	no
2	F.B.	forearm middle 1/3	left	5 years	trauma	standard	no
3	M.G.	forearm middle 1/3	right	22 years	trauma	standard	no
4	E.C.	forearm distal 1/3	right	3 years	infection	standard/CONM	no
5	S.B.	forearm distal 1/3	right	8 months	trauma	standard	yes
6	V.R.	midcarpal	right	12 years	trauma	standard	no

**Table 2 bioengineering-10-00319-t002:** Characteristics of the test subjects.

Subject No.	Subject	Age (Years)	Sex	Height (cm)	Weight	Amputation Level	Unilateral/Bilateral/No Amputation
1	F.B.	32	M	179	86	forearm/forearm	bilateral
2	M.G.	59	M	171	78	forearm	unilateral
3	S.B.	35	M	178	82	forearm	unilateral
4	V.R.	47	M	174	77	midcarpal	unilateral
5	E.C.	37	M	176	74	forearm	unilateral
6	R.I.	32	M	174	69	no amputation	no amputation

**Table 3 bioengineering-10-00319-t003:** Anatomical characteristics of the amputation stumps.

Criteria	1	2	3	4	5	6
Circumference proximal 1/3 in mm	240	267	253	250	248	249
Circumference middle 1/3 in mm	137	176	173	165	-	198
Circumference distal 1/3 in mm	-	120	113	-	-	114
Fat layer thickness proximal 1/3 (mm)	11	12	9	10	12	12
Fat layer thickness middle 1/3 (mm)	11	8	10	9	-	10
Fat layer thickness distal 1/3 (mm)	-	4	3	-	-	3
Flexion deficit—elbow	<5°	<5°	<5°	<5°	<5°	<5°
Extension deficit—elbow	<5°	<5°	<5°	<5°	<5°	<5°
Prono-supination deficit	<5°	<5°	<5°	<5°	<5°	<5°

**Table 4 bioengineering-10-00319-t004:** Anatomical characteristics of healthy forearms.

Criteria	1	2	3	4	5	6
Circumference proximal 1/3 in mm	280	323	277	275	312	256
Circumference middle 1/3 in mm	224	240	214	210	254	210
Circumference distal 1/3 in mm	161	165	142	141	173	154
Fat layer thickness proximal 1/3 (mm)	12	14	12	11	13	14
Fat layer thickness middle 1/3 (mm)	12	10	11	10	-	12
Fat layer thickness distal 1/3 (mm)	-	3	4	-	-	3
Flexion deficit—elbow (degrees)	<5°	<5°	<5°	<5°	<5°	<5°
Extension deficit—elbow (degrees)	<5°	<5°	<5°	<5°	<5°	<5°
Prono-supination deficit (degrees)	<5°	<5°	<5°	<5°	<5°	<5°

**Table 5 bioengineering-10-00319-t005:** Mean amplitude values for sEMG signals emitted by the flexor carpi radialis, brahioradialis and extensor digitorum communis muscles.

	Case No.	Healthy Forearm	Amputation Stump
Mean amplitude values for sEMG signals emitted by the flexor carpi radialis muscle	1	4,6 nW	0.043 nW
	2	4.0 nW	0.037 nW
	3	3.9 nW	0.047 nW
	4	4.3 nW	0.062 nW
	5	3.6 nW	0.034 nW
	6	4.8 nW	0.046 nW
Mean amplitude values for sEMG signals emitted by the brahioradialis muscle	1	3.8 nW	0.048 nW
	2	3.5 nW	0.027 nW
	3	4.5 nW	0.043 nW
	4	2.8 nW	0.038 nW
	5	3.4 nW	0.042 nW
	6	2.9 nW	0.047 nW
Mean amplitude values for sEMG signals emitted by the extensor digitorum communis muscle.	1	4.3 nW	0.043 nW
	2	4.8 nW	0.047 nW
	3	3.2 nW	0.039 nW
	4	4.6 nW	0.055 nW
	5	3.6 nW	0.041 nW
	6	4.5 nW	0.038 nW

## Data Availability

All data can be obtained via e-mail from the corresponding author.

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
