# Peer review of "The Role of Fascial Tissue Layer in Electric Signal Transmission from the Forearm Musculature to the Cutaneous Layer as a Possibility for Increased Signal Strength in Myoelectric Forearm Exoprosthesis Development"

_bioengineering, 2023, doi:10.3390/bioengineering10030319_

Round 1

Reviewer 1 Report

Dear Editor,

This is an interesting. the authors investigated The role of fascial tissue layer in electric signal transmission from the forearm musculature to the cutaneous layer as a possibility for increased signal strength in myoelectric forearm exo-prosthesis development. There are some comments which are listed below:

- Introduction is so long 

-Method and materials need to be categorized as subtitles 

-Results need to be categorized as subtitles 

-Discussion needs more new references ( 2015-2023)

Good Luck

Author Response

Thank you for the prompt review. We managed to implement all of your suggestions:

(1) The Introduction section is now much shorter and concise. It now has 690 words instead of 1390;

(2) Materials and Methods now has been subdivided by subtitles to better highlight the content (5 subtitles have been added);

(3) Results section has also been subdivided in subtitles: 3.1 to 3.4;

(4) We expanded the discussion section with more references taken into account, with newer titles. Now the manuscript has 59 citations instead of the original 26.

Reviewer 2 Report

Overall comments

The study addresses an important issue. However, the authors have failed to provide a clear purpose to the study. The methods are given in detail, but not in a logical order and they appear chaotic. Due to the absence of a clear (or specific) purpose (or purposes), the discussion seems to lack a logical succession of the interpretation of the results. The authors need to make major revision to their interesting study before it can be considered eligible for publication.

Specific remarks

Abstract

line 36: correct to "the role of the forearm fascial layer..."

line 46: correct to :...in the control of modern exoprosthetics".

Introduction

line 50: it is suggested to write "psychological demise" in order not to repeat the word

line 51: it is suggested to delete "...and changes..."

line 66: correct to "...obtain clearer and more distinct..."

line 74: correct to "...can measure..."

lines 94-95: it is suggested to rephrase like "Furthermore, the intensity of the sEMG signal can be significantly affected by the movement of each muscle".

line 96: correct to "as well as scar..."

lines 99-104: what is the relevance of this paragraph with the study's population? it is general knowledge of the EMG method function that could be applied to either healthy or non healthly subjects. The authors are advised to relate the use of surface emg sensors with exoprosthesis in the previous paragraph with a short phrase.

lines 126-141: this paragraph reports on 2 studies that examined alternative methods to address all the issues arising from using conventional sEMG signals. The study by Culjat et al (2010) preceded the one by Pasquina et al (2015), which in my opinion, is closer to the current study's purpose. therefore, it is suggested that the paragraph is re-written in an opposite order of appearance -first the one by Culjat et al (2010).

 Further, I suggest that the authors place this (corrected) paragraph first and then proceed with explaining what does the EMG method do and its particularities with amputated limbs. So, the 3 previous paragraphs (lines 99-105, 106-114 and 115-125) are suggested to be integrated in one, more compact paragraph.

lines 142-156: the introduction should arrive at the formulation of the study's purpose. It is recommended that the authors make more clear the relevance of the information along those lines with the purpose of their study.

Materials and methods

General comment: It is highly recommended that this section is divided in sub-sections, for example Sample (or Participants) etc. Also, please refer to my overall comment about the Methods above.

lines 165-171: as suggested earlier, these lines should be moved to and integrated with the last two paragraphs of the introduction in order to provide readers with a clear and meaninful purpose.

line 172: the authors need to briefly describe the recruitment process. 

lines 182-188: it is suggested that these lines are re-written, starting with something like "The test subjects were assigned to two groups. Group 1 comprised of 6 amputated forearms and Group 2 with 6 non-amputated forearms, specifically...."

Lines 189-209: a) measurement of anthropometric and kinematic data (lines 189-194)

b) recording of EMG data (lines 195-199 and 210-211), along with skin preparation for both the amputated and non-amputated limbs (lines 203-209). 

c) data acquisition and processing of the EMG signal = lines 212-217, 230-239.

it is recommended that those lines are given appropriate sub-headings.

Lines 199-202: it is not clear what do they authors want to say here. if these lines refer to specific instructions with regards to the voluntary isometric contractions performed, then it is advised that they join lines 210-211. Otherwise, the authors need to clarify.

line 217: please clarify what does the effective contraction intervals refer to. 

Lines 219-220. Figure 1: in fig1 (a) if the purpose is to show the software used for EMG data recording, then the photo should focus on the screen showing the software's interface.

In fig1(b), the photo should be replaced with another one showing from a closer range the electrodes that are placed in the patient's amputated limb.

Lines 221-225: If understood correctly, these lines refer to the muscles that underwent training since they are forearm muscles. However, either these lines need to be integrated somewhere else or the authors need to re-write them in a way to show relevance with recording of the EMG data.

Paragraphs in:  1st) lines 243-253, 2nd) 254-264, 3rd) lines 268-283, 4th) lines 288-296, 5th) 297-306, 6th) 307-317, 7th) 318-326

As afore-mentioned, sub-headings need to be added and the order of those paragraphs be re-arranged. It is very difficult to follow the manuscript without a logic sequence of the presentation of information. 

Lines 284-287: it is suggested that Fig.3(b) and (c) are joined together in order to show movement execution from static position to active movement. However, may be the authors want to give another information with fig.3.b, so please clarify.

Results

It is recommended that figures 4-5-6 are removed and that tables 5-6-7 are integrated into one table. the use of both figures and tables with the same information is unnecessary.

Lines 339-341: this is a repetition of  the data analysis process. It is suggested that it is re-written in a way such as "Tables 5 to 7 present the results on the sEMG signal amplitude of flexor carpi radialis ...muscles over the effective contraction interval".

lines 356-358: it is recommended to keep Figure 7 and discard Table 8 for this case.

Discussion

line 391: correct to observed instead of noted.

line 399: correct to "found" from "find".

Line 405: the authors write about "Group 1 having a significantly decreased amplitude when compared to Group 2" however, they have not reported any statistical method of testing their hypothesis in the Methods. They need to address this issue.

Lines 457-464: this whole phrase needs to be reduced down to 3 phrases; it is way too long.

Lines 474-477: the relevance of fascia tissue with the purpose of the study eludes me, please clarify this. 

Limitations

Lines 479-480: the authors argue that due to small sample size, statistical testing was not performed, however in the discussion (line 405) they wrote about statistical significance. The issue of statistical testing needs to be addressed, may be with a correction for very small sample sizes.

Conclusions

The Conclusions need to be re-written in a shorter, more compact way. This part of the manuscript reads more like a continuation of the discussion rather than concluding remarks of the study.

Author Response

Dear reviewer, thank you for your detailed review and feedback.

We managed to implement most of your requests and suggestions:

Abstract:

- line 36: corrected 

- line 46: corrected 

Introduction

- line 50: corrected

- line 51: corrected

- line 66: corrected

- line 74: corrected

- lines 126-141 - now the proper timeline in citations has been restored.

- lines 94-95: corrected

- line 96: corrected

- lines 99-104: - completely rewritten, to better reflect your indications

- lines 142-156: the introduction should arrive at the formulation of the study's purpose. - the paragraph has been rewritten

Materials and Methods

- General comment: - we added subtitles in this section to better highlight the scientific content

- lines 165-171: corrected

- observations regarding (line 172), (lines 182-188), (lines 189-209), (lines 199-202) - all have been addressed by rewriting the paragraphs.

- line 217: rewritten in order to clarify

- lines 219-220 - Fig 1 has been modified accordingly

- Paragraphs in:  1st) lines 243-253, 2nd) 254-264, 3rd) lines 268-283, 4th) lines 288-296, 5th) 297-306, 6th) 307-317, 7th) 318-326 - subheadings have been added

- Lines 284-287: Fig.3 (b) and (c) - Pictures in (b) and (c) have been merged into a single image, in (b). Also the caption has been changed to reflect this.

Results

- It is recommended that figures 4-5-6 are removed and that tables 5-6-7 are integrated into one table. the use of both figures and tables with the same information is unnecessary. - Done

- Lines 339-341: - Done.

- lines 356-358: it is recommended to keep Figure 7 and discard Table 8 for this case - Done. 

Discussions

- line 391: correct to observed instead of noted. - Done

- line 399: correct to "found" from "find". - Done

- Line 405: - PENDING

- Lines 457-464: this whole phrase needs to be reduced down to 3 phrases; it is way too long. - Done

- Lines 474-477: - revised

Limitations

- Lines 479-480: this has been addressed in the secions above.

Conclusions:

- This section has been completely rewritten in a much shorter manner.

Reviewer 3 Report

Thank you for permitting me to review the manuscript

This paper claim describing the role fascial tissue layer in electric signal transmission in the abstract, title  , while in the method section the main and secondary objective are different (reverted) , please be consistent

Here are my suggestions

Abstract : should describe better the results in particular the difference in signal ratio between healthy and amputated forearm

Line 120-130 please provide some additional graphic to permit the reader to better understand the electrode disposition

Some graphic representation of the signals would help if possible

Please better describe the AcqKnow software in addition add more details about the company

Please delete figure 4-5-6 7 as they are unnecessary

                Fig 9 nothing is visible

The results give too much importance to the chronic pain patient , it should be shortened in this section and elaborated for the other patients

Line 485 : first paragraph this result are rather expected

Line 492, If the results are not significant, speculation should be cautious

Is there any other perspectives in the absence of fascia layer

The small number of patients describe rather a case series not a pilot study

Author Response

Dear reviewer, thank you for your detailed review and feedback.

We managed to implement most of your requests and suggestions:

- abstract: has been rewritten, to better reflect the results of the study.

- Some graphic representation of the signals would help if possible: we added a new image (Fig. 3 - d), a direct export of the software interface used for signal analysis.

- Please better describe the AcqKnow software in addition add more details about the company: we added the apropriate information

- Please delete figure 4-5-6 7 as they are unnecessary - Done

- Fig 9 nothing is visible - In the version of the manuscript that we have uploaded, Figure 9 has 2 pictures that diplay correctly. If you need these pictures to be sent separately, please let us know.

- Line 485 : first paragraph this result are rather expected: we modified the conclusions section to better reflect the results and complement the discussions.

- Line 492, If the results are not significant, speculation should be cautious: We modified the conclusions.

- The small number of patients describe rather a case series not a pilot study - We never mentioned that this manuscript relates to a pilot study, but rather a limited study.

Round 2

Reviewer 2 Report

Dear authors,

Your revised manuscript is significantly improved. Congratulations. The length of the discussion serves, in my opinion, as relatively discouraging, however I have no suggestions at this point.

One minor correction is required in lines 378-380, where some words are needed so that the phrase makes sense.

Also, line 378: correct "Jiang et all" to Jiang et al

Line 383: correct "mai" to may

Reviewer 3 Report

The authors have adequately responded to my queries